# Sleep Pathologies and Eating Disorders: A Crossroad for Neurology, Psychiatry and Nutrition

**DOI:** 10.3390/nu15204488

**Published:** 2023-10-23

**Authors:** Carlotta Mutti, Giulia Malagutti, Valentina Maraglino, Francesco Misirocchi, Alessandro Zilioli, Francesco Rausa, Silvia Pizzarotti, Marco Spallazzi, Ivana Rosenzweig, Liborio Parrino

**Affiliations:** 1Sleep Disorders Center, Department of General and Specialized Medicine, University Hospital of Parma, 43125 Parma, Italy; 2Neurology Unit, Department of Medicine and Surgery, University of Parma, 43125 Parma, Italyalessandro.zilioli@unipr.it (A.Z.);; 3Plasticity Centre, Department of Neuroimaging, Institute of Psychiatry, Psychology and Neuroscience (IoPPN), King’s College London, London WC2R 2LS, UK

**Keywords:** nutrition, sleep disorders, sleep, eating habits, eating disorders, metabolism

## Abstract

The intricate connection between eating behaviors and sleep habits is often overlooked in clinical practice, despite their profound interdependence. Sleep plays a key role in modulating psychological, hormonal and metabolic balance and exerting an influence on food choices. Conversely, various eating disorders may affect sleep continuity, sometimes promoting the development of sleep pathologies. Neurologists, nutritionists and psychiatrists tend to focus on these issues separately, resulting in a failure to recognize the full extent of the clinical conditions. This detrimental separation can lead to underestimation, misdiagnosis and inappropriate therapeutic interventions. In this review, we aim to provide a comprehensive understanding of the tangled relationship between sleep, sleep pathologies and eating disorders, by incorporating the perspective of sleep experts, psychologists and psychiatrists. Our goal is to identify a practical crossroad integrating the expertise of all the involved specialists.

## 1. Introduction

Since the early 1800s, the concept of a profound connection between men’s dietary habits and their overall health has steadily gained recognition. In 1826, in an essay named in *Physiologie du Gout*, Anthelme Brillat-Savarin wrote “*Dis-moi ce que tu manges, je te dirai ce que tu es*”-alias ‘*Tell me what you eat, and I will tell you what you are*’. A few years after, Ludwig Andreas Feuerbach declared “Der Mensch ist, was er ißt.”-Man is what he eats.

It has been theorized that the discovery of fire and the subsequent development of ‘cooking habits’ exerted a strong evolutionary effect. According to the Wrangham hypothesis, the shift from eating raw foods to preparing cooked meals allowed humans to spend less energy for digestion and chewing, reducing the need for a long digestive tract and, conversely, permitted the growth of brain structures, requiring higher rate of metabolic energy [1]. In parallel, the possibility of being sheltered by fire allowed our ancestors to adapt to ground-based life, as they were finally protected against predators. Moving from arboreal to terrestrial environments, human sleep became shorter, deeper and increased the proportion of rapid eye movement (REM) sleep periods [2].

Less sleep enabled longer active periods to acquire and transmit new skills and knowledge, slow wave sleep was critical for the consolidation of those skills, while enhanced REM sleep led to enhanced cognitive abilities in early humans [3,4].

Although the role of cooking has been considered neither sufficient nor necessary to explain brain expansion during evolution [5], humans have made astonishing progress since our ancestors came down from the trees [6] and the fascinating relation connecting environment, food and sleep warrants major attention. Nutrition needs are deeply influenced by social status. Hunter gatherers are far less interested in lives of food abundance compared to agricultural communities [7].

As individuals migrate to urban areas, characterized by an abundance of food stores and supermarkets, malnutrition becomes a prevalent issue among low-income residents who often face limited access to nutritious food options [8]. It is known that unhealthy eating habits are considered among the major risk factors for a wide range of chronic diseases, such as diabetes, hypertension, cardiovascular diseases and cancer. The correlation is so strong that the attainment of healthy diets is prominently featured as one of the foremost priorities in the latest campaigns by the World Health Organization (WHO) [9]. Of equal importance are untreated sleep disorders that frequently lead to excessive daytime sleepiness (favoring accidents and car crashes), with indirect economic and professional consequences and may also yield to severe metabolic disturbances, obesity, augmented cardiovascular risk, neuropsychiatric disorders and tumors [10].

Despite both sleep and eating disorders being independently linked to various chronic conditions, there is currently a lack of emphasis on investigating their reciprocal influence on one another. Food industry, social media and advertising exert a powerful influence on social habits and eating behaviors from early childhood. Similarly, routine utilization of electronic devices can favor a delay shift in circadian sleep–wake rhythms, augmenting sleep latency and, sometimes, fragmenting sleep quality and amount [11].

Education, family habits and genetic background might influence both sleep–wake cycle and eating behaviors. The relationship is even more complex in the scenario of sleep and eating pathologies, when unhealthy trends might exacerbate pre-existing frailties, increasing the clinical burden in affected patients. Eating disorders are common among patients affected by sleep disorders and, in turn, sleep disorders may worsen the clinical course of patients with eating disorders. Diet and sleep reciprocally mutually exert influence on each other in multiple ways, notably through undisclosed or subtle psychological and behavioral factors. In this review, we summarize the key aspects of the intermingled relationship between sleep disorders and eating disorders, combining a clinical and psychological perspective in the modern life scenario.

## 2. Eating Disorders and Sleep: What the Sleep Expert Should Know

Unhealthy eating behaviors and sleeping habits frequently depend on similar somatopsychic and psychosomatic etiological factors. Extensive evidence has already demonstrated the involvement of various psychological aspects in both the development and chronicity of eating disorders (EDs) [12]. Similarly, personality traits, anxiety and depression might predispose and perpetuate the development of various sleep disorders, including, above all, chronic insomnia [13]. Patients affected by an ED frequently present an obsession with food, eating habits, body weight and/or body shape that, if untreated, might lead to life-threatening consequences. Moderate-to-severe depressive symptoms are frequently observed in patients affected by EDs, with anxiety and affective disorders being the commonest comorbidities [12,14,15,16,17]. Notably, depression and anxiety has been suggested as one of the major determinants of the link between insomnia and EDs [16].

Given their pathogenetic commonalities, it is not surprising that EDs and sleep disorders frequently coexist. This epidemiological overlap between sleep disturbances and EDs has been also attributed to the disruption of sleep-dependent hormonal mechanisms, including variations in orexin, cortisol, ghrelin and leptin signaling [18]. The latest years faced a steep increase in the incidence of ED: overall female sex is more affected and diseases onset take place usually during adolescence. Epidemiologically, eating disorders (EDs) are most prevalent in Western countries, with a particularly high incidence in the United States [17]. This is likely attributed to a combination of cultural influences, lifestyle habits, genetic predispositions, and variations in gut microbiota.

### 2.1. Anorexia Nervosa (AN)

Anorexia nervosa (AN) is one of the most common and potentially life-threatening ED. It is associated with food restriction, fear of weight gain and BMI < 18.5 [19]. Although less common among males, there has been a significant increase in its prevalence within this population in recent years [19,20]. Males affected by AN are commonly identified as ‘atypical’ AN cases, as they sometimes do not fulfill current diagnostic criteria for AN, overcomplicating the diagnostic process. Instead of being worried about their weight, anorexic males usually present more ‘muscularity concerns’ and body dissatisfaction. Also, they tend to over-exercise to increase their muscle mass compared to AN females, who focus on leanness-oriented exercises [21]. These gender-related differences in the AN phenotype likely depend on multiple factors (e.g., hormonal, genetic, socio-cultural and psychological differences) but both male and female anorexic patients warrant a multidisciplinary work-up. A recent research paper adopting a network-based analysis to conceptualize core features of AN in a cohort of 267 patients (94.8% female) focused on the central role of sleep disturbances in the disease pathophysiology [22]. According to this study’s results, both sleep and anxiety strongly impact numerous AN’s symptoms and, therefore, clinicians should always try to target them to improve AN management. Furthermore, sleep plays a crucial role in modulating the synthesis and concentration of ghrelin and leptin, hormones involved in food intake and body weight regulation, which can increase individuals’ vulnerability to obesity and eating disorders. Anorexic patients can experience initial insomnia, restlessness, increased awakening and non-refreshing sleep, but also delayed sleep–wake circadian rhythm disorders [23]. Remarkably, sleep characteristics appear to exhibit a certain parallelism with the severity of the disease. Within this context, the progressive weight restoration observed in treated individuals with anorexia nervosa (AN) is commonly accompanied by an augmentation in total sleep duration and the percentage of REM sleep [24]. Consequently, assessing sleep patterns in AN patients may hold prognostic significance, as improvements in sleep could serve as a confirming indicator of the effectiveness of the ongoing treatment.

### 2.2. Bulimia Nervosa (BN)

Bulimia nervosa (BN) is an ED characterized by binge eating followed by purging. The link between sleep disturbances and BN is less clear-cut and bulimic patients hardly report insomnia symptoms. An association of BN with disrupted circadian rhythm disruption had been suggested, especially among patients experiencing binge eating) in the late evening/at night, as eating represents a chrono regulatory clue favoring wakefulness maintenance [25]. Historical observations also pointed out an association between sleepwalking and BN, suggesting a potential link between the two conditions [26]. Notably, at the polysomnographic level, a correlation between the severity of the disease and the percentage of lighter sleep stages (specifically stage N1 of NREM) has been suggested, while no consistent association has been revealed between REM sleep and depression symptoms in BN [27]. The neurobiological significance of changes in REM sleep and NREM sleep in different subtypes of ED require further investigation.

### 2.3. Binge Eating Disorder (BED)

Binge eating disorder (BED) is another extremely common ED, highly comorbid with obesity, and distinct by repeated episodes of compulsive, non-homeostatic food consumption and associated with the perception of loss of control over how much is consumed [19]. Unlike BN, BED is typically not followed by compensatory behaviors (e.g., vomiting, abuse of laxatives, excess of exercise). In many cases, patients affected by BED use food and overeating to cope with masked depressive symptoms or anxieties. Unrefreshing sleep and chronic insomnia are extremely frequent among patients affected by BED [28,29,30]. A proper sleep evaluation in patients affected by ED is overcomplicated by their tendency to mask most of their symptoms [12]. For example, AN patients commonly exhibit strong perfectionist traits and could take advantage of the time ‘steal’ to their physiological sleep for other high-level performances. As a consequence, they rarely complain about either insomnia or sleep disturbances. Conversely, patients affected by BN and BED might experience night compulsive overeating, which may favor insomnia development and circadian rhythm delay, but they frequently live these compulsive behaviors with shame and sense of inadequacy, eventually hiding them at clinical evaluation [12].

From the instrumental perspective, polysomnographic (PSG) sleep analysis in patients affected by ED revealed poor sleep quality, high amount of awakening and, at microstructural level, a pathological increase of cyclic alternating pattern (CAP) oscillations, an electrophysiological index of sleep stability [31]. However, instrumental data are extremely scarce, as PSG recording is not routinely performed in patients affected by ED. Interestingly, some of the key psychopathological processes involved in the development of ED (e.g., anxiety, low self-esteem, perfectionism, obsession for food and body shape), sometimes reveal themselves into patients’ dream content: ED patients can experience inadequacy, anger, regret or self-hate while dreaming and typically refer lower amount of dreaming recall compared to age-matched healthy controls [31,32,33]. More specifically, the dreams of individuals with BN typically revolve around concerns related to food and nutrition. Additionally, there is often a notable presence of hostility towards other individuals featured in the dream content [34]. On the other hand, AN patients’ dreams exhibit distinct characteristics with reduced levels of anger and hostility compared to BN patients. Instead, they are predominantly dominated by anxiety and portray negative impending scenarios [29,30]. Another fascinating hypothesis trying to explain the link between ED and sleeping disorders focuses on the role of the orexinergic system [35]. Orexin is one of the key regulators of arousal, wakefulness and appetite/food habits; in parallel, orexin peptides also influence autonomic and metabolic balance, reward and motivation behaviors [36]. Regarding the latter, the orexinergic signaling system has been implicated in regulating homeostatic feeding behaviors, particularly following periods of fasting. Orexin release increases towards salient stimuli, such as highly palatable foods (or drugs of abuse), hence chronic binge eaters alternating overeating and fasting may favor the increase the excitability of the orexinergic system, favoring sleep disturbances development and a ‘hyperarousal’ state [35] (Figure 1). This intriguing hypothesis could potentially have therapeutic implications for ED. Recently, there have been advancements in the development of orexin receptor antagonists primarily for managing insomnia and, theoretically, these antagonists might also prove beneficial in the treatment of ED [37].

### 2.4. Obesity

Obesity is defined as abnormal or excessive fat accumulation that presents high risk to general health. A body mass index (BMI) over 30 is considered obese and nowadays the condition, historically associated to high-income countries, is now dramatically on the rise in low- and middle-income countries, particularly in the urban settings, with even higher rate among children according to World Health Organization data.

Although traditionally considered a uniform condition, it is now well-known that among obese patients, numerous heterogenous phenotypes can be recognized, depending on the participation of molecules, genes and cells, in addition to environmental, social and economic factors or coexistent comorbidities [38].

Shorter sleep duration has been associated to higher risk for overweight and obesity development, likely due to the increased risk for hedonic eating behaviors in sleep restricted subjects, which also appear to have more opportunities to eat as they stay awake for more hours in the day [39]. Shorter sleep duration promotes various hormonal changes associated with appetite regulation, specifically leptin and ghrelin secretion [40]. Besides sleep duration, also eating timing, which impacts the dynamic changes of the internal clocks, has been linked to obesity risk, especially among younger generations. Maintaining a regular and early bedtime schedule during the week, as well as across weekdays and weekends is considered a key aspect to prevent the development of obesity intoddlers [41].

Obese patients are at risk for numerous sleep pathologies including, above all, sleep-related breathing disorders such as obstructive sleep apnea (OSA) and obesity hypoventilation syndrome (OHS). The excess in body weight might directly compress the upper airways, favoring their repetitive partial collapse during the night, which can hamper the physiologic diaphragmatic excursion and reduce lung capacity. The relationship between OSA and obesity is so tight that bariatric surgery is considered one of the key treatments for this sleep-related breathing disorder [42]. Both OSA and OHS increase the burden for cardiovascular, metabolic, mood and cognitive disturbances in obese subjects.

Excessive daytime sleepiness (EDS) and fatigue had always been considered as directly linked to sleep-related breathing disorders in obese patients; however, recent evidence suggests that rather than being directly associated to concomitant sleep pathologies, these highly prevalent symptoms in obese patients may reflect a complex interaction between the hypothalamic-pituitary-adrenal (HPA) axis and proinflammatory cytokines (especially TNF alpha and interleukin-6) with the circadian system. The level of inflammatory cytokines is largely mediated by the severity of obesity (expressed as BMI), while the role of HPA axis is more variable among obese subjects [43]. Schematically, it has been proposed that the condition of hypercortisolemia (plus hypercytokinemia) might be associated with low sleep efficiency and fatigue, while either the eucortisolemia or the hypocortisolemia (plus hypercytokinemia) can determine a better sleep efficiency with objective sleepiness [44]. In this perspective, EDS could be considered a direct consequence of metabolic and pro-inflammatory symptoms of obesity itself, rather being related to concomitant sleep pathologies.

The scenario becomes even further complicated when obesity overlaps with other neuroendocrine disturbances, such as in the case with Prader–Willi syndrome (PWS). PWS is a rare complex disorder related to the loss of expression of paternal chromosome 15q11.2-q13 and is associated particularly with the deletion of the small nucleolar ribonucleic acid-116 (*SNORD116*). SNORD116 modulates the sleep–wake pattern, which influences the orexin-hormone system and modulates feeding habits [45]. PWS is typically associated with hyperphagia and obesity, and it is strongly linked to numerous sleep-related pathologies including central and obstructive sleep apnea, hypersomnia, narcolepsy-like phenotypes and insomnia, whose management should become an integral part of the medical approach [46].

According to these premises, obese patients might benefit from a sleep evaluation in order to rule out the existence of sleep pathologies which may aggravate their clinical status. Similarly, patients with short sleep duration and/or lower sleep quality should be monitored for their higher risk for unhealthy eating pattern and obesity.

## 3. Sleep-Related Eating Behaviors: What the Psychiatric Should Know

### 3.1. Nocturnal Eating Syndrome

Nocturnal eating syndrome (NES) is described as a conscious and abnormal eating behavior with evening hyperphagia consuming >25% calorie intake and/or nocturnal awakening with food ingestion and morning anorexia, occurring ≥2 times per week [19]. Patients with NES typically experience a compulsive desire to eat at night and may also believe that eating is necessary to sleep [47]. Mean length of NES episodes is estimated to be around 3.5 min, with ‘eating latency’ (time between the awakening and the beginning of the eating behavior) of less than 1 min in ~50% of cases [48]. Night eaters frequently complain of lower sleep quality but do not suffer from excessive daytime sleepiness [49]. Fragmented sleep, with multiple praecox nocturnal awakenings after sleep onset may be appreciated with objective sleep recording [50] and longitudinal studies reported higher risk for obesity, metabolic syndrome and dyslipidemia among untreated patients, with higher odds among women [51].

The true prevalence of this condition is probably largely underrated, as both clinicians and patients do not always explore the topic during regular medical visits. Additionally, it is worth noting that patients with nocturnal eating may often underreport their nighttime eating episodes. This underreporting can stem from feelings of shame or embarrassment surrounding their behavior. Moreover, there is a misconception that nocturnal eating is merely a habit rather than a clinically significant disturbance.

Reasonably, patients with ‘evening-type’ eating habits would probably refer to psychiatric centers or eating-disorder centers, while those with ‘nocturnal-type’ eating behaviors may prefer a sleep-center for consultation. However, it is important to recognize that in many cases, individuals with these eating behaviors may experience distress, symptoms of compulsive disorders and other sleep-related disturbances. As a result, it is crucial for both sleep experts and mental health professionals in psychiatry to be aware of this condition and its multifaceted consequences. Indeed, the development of nocturnal eating syndrome (NES) is likely influenced by a combination of genetic predisposition and external or acquired factors. Overall, NES is characterized by a significant delay in the circadian rhythm of food/drink intake associated with a corresponding delay in numerous neuroendocrine functions (leptin, melatonin, prolactin, cortisol and thyroid stimulating hormone) [52].

#### 3.1.1. NES Management

NES management is typically based on a combination of pharmacological and non-pharmacological treatments. Drugs adopted include antidepressants such as sertraline, paroxetine, venlafaxine and escitalopram [53,54], as a reduction of post-synaptic serotonin availability had been theorized in the disease pathogenesis. Their effectiveness however varies according to trials, and a relevant amount of patients discontinued therapies due to medication side-effects (such as nausea, diarrhea, dyspepsia, hepatotoxicity, weight gain, urinary retention, incontinence, sexual dysfunction, cognitive disturbances, affective disturbances, glaucoma and others).

Given the hypothesized association of NES with circadian sleep disturbances, some chronobiological interventions had also been tested, including melatonin supplementation [55]. Few patients tried agomelatine with promising results (improvement in symptoms, weight loss and favorable metabolic profile) [56]. However, the significant risk for acute iatrogenic hepatotoxicity limits its feasibility.

Topiramate, an antiepileptic agonist of gamma-aminobutyric acid influencing taste perception with mild sedative effect, is sometimes adopted for NES management. Caution should be used when treating psychiatric patients for the potential neuropsychiatric side effects of this medication [57]. Non-pharmacological interventions for NES include cognitive-behavioral therapy (CBT), phototherapy (morning exposition to bright light) and muscle relaxation, all ameliorating patients’ perceived stress, supporting the role of education in this nocturnal disorder management [58,59,60]. In particular, psychosocial and supportive interventions (performed individually or with the involvement of patient’s relatives), represent a valuable help for a comprehensive NES management [61].

#### 3.1.2. NES and Coexistent Sleep Disorders

NES is commonly observed among patients with various sleep disorders, including insomnia, restless leg syndrome (RLS), periodic limb movement disorder (PLMD) and obstructive sleep apnea (OSA).

RLS is described by an urge-to-move or a discomfort/unpleasant sensation, predominantly in the evening, worsened with inactivity and localized to the lower extremities, that may interfere with sleep onset and quality [62]. Patients with RLS may sometimes be affected by other non-motor disturbances, such as compulsive eating behaviors and/or NES. These behaviors may manifest during late evening, as sometimes the craving for food is lived by the patients as an urgent ‘need to solve’ in order to fall asleep immediately afterwards or, in other cases, the compulsive eating behavior may appear only after first nocturnal awakenings.

Interestingly, NES and RLS share a common circadian distribution, with a peak in late evening and a progressive decline overnight: this overnight/ultradian pattern could at least partly be correlated with the circadian rhythm of core body temperature, melatonin dynamics and dopamine activity [63,64,65].

Furthermore, patients with NES often exhibit a misalignment of their circadian rhythm, displaying a tendency towards evenings [47]. Managing patients with both RLS and NES can be challenging. Serotonin reuptake inhibitor (SSRI) drugs like sertraline and escitalopram, commonly used for psychiatric conditions, can worsen RLS symptoms [53,60]. Conversely, dopaminergic drugs such as pramipexole and ropinirole are effective treatments for RLS, but may not be ideal for patients with concomitant compulsive disorders like NES.

The relationship between NES and OSA is not straightforward, as explorative studies have produced partially divergent results [66,67,68]. Generally, the frequency of sleep-related eating disorders is not directly associated with the severity of OSA, measured by the apnea/hypopnea index (AHI). However, OSA and NES can coexist in patients with psychiatric comorbidities [67]. It is worth noting that dopaminergic medications used for conditions like Parkinson’s disease (PD) can trigger compulsive nocturnal binge eating, which can be reversed by discontinuing medication [68].

### 3.2. Sleep-Related Eating Disorder (SRED)

Sleep-related eating disorder (SRED) is defined by recurrent episodes of dysfunctional eating at the transition from nighttime sleep to arousal, typically associated with reduced consciousness. Consumption of non-edible/toxic food and potentially injurious behaviors may be observed, with various adverse health consequences eventually possible [62]. SRED may appear isolated or associated with other sleep-disorders, such as NREM-sleep parasomnia, RLS and OSA. Iatrogenic SRED, mostly due to the benzodiazepine receptor agonist zolpidem or associated with tricyclic antidepressants, anticholinergics, lithium, antipsychotic medications and sodium oxybate have also been described [69,70,71]. SRED may also overlap with other eating disorders such as AN e BN, with estimated prevalence of around 5%, often associated with depression and dissociation feelings [70]. The major difference between NES and SRED is the level of consciousness during the episode of nocturnal eating, which is lower in SRED (sleepwalking-like behavior with only partial recall or total amnesia for the episode). Patients with SRED appear confused and sometimes alarmed by the episode of nocturnal eating, and they commonly realize only after awakening (Figure 2). The association between SRED and NREM parasomnia is so tight that the former had been included as parasomnia itself [62]. As for all NREM sleep parasomnia, various factors can contribute to increased sleep fragmentation or, conversely, deepening of sleep, thereby favoring SRED occurrence. NREM parasomnias are deemed to be arousal-related phenomena strongly associated with incomplete arousal from sleep. Accordingly, sleeping brain areas coexist with awakened brain areas during NREM parasomnia episodes. All factors raising the level of sleep instability (such as coexistent untreated sleep disorders and/or external disturbing factors) will precipitate SRED relapse.

#### SRED and Coexistent Sleep Disorders

SRED is frequently reported as a medication side-effect, even if, at least in some cases, a genetic predisposition to this abnormal nocturnal behavior may be identified [72]. Zolpidem utilization is recognized as one of the major risk factors for SRED occurrence. Zolpidem is short-acting Z-drug, with low abuse potential, highly selective towards alpha-1 subunits in GABA-A receptors [73]. It significantly shortens sleep latency, being a reasonable choice to treat initial insomnia patients [74]. Zolpidem effects on sleep dynamics are multiple and include the reduction of EEG power across most frequencies during NREM sleep, prolongation of NREM sleep and mild REM sleep reduction [75]. A recently published literature review explored SRED features associated with zolpidem utilization [76]: among the 40 analyzed cases, 65% had concomitant sleep disorders including OSA or RLS and/or were affected by depression. A higher risk for SRED was evident among patients using higher daily doses (≥10 mg), and/or with concurrent use of antidepressants or other psychiatric medications. SRED has indeed been rarely described in patients with PD, particularly when coexistent untreated sleep disturbances (OSA, PLM or REM-sleep behavior disorder—RBD) leading to sleep fragmentation exist [77] or, anecdotally, as side effect of pramipexole utilization [78]. Atypical antipsychotic drugs and mirtazapine had also been associated with SRED episodes [79,80,81]. SRED can potentially lead to the unconscious ingestion of toxic substances, making it crucial for sleep physicians to make every effort to identify and eliminate any triggers in affected patients. Managing SRED can be challenging, and there are a few treatment options that have shown promise. Some evidence suggests that bupropion, an antidepressant with dopaminergic and noradrenergic effects, may be effective in reducing abnormal nocturnal eating in patients with alcohol and/or drug abuse issues [81,82]. Another medication, topiramate, has demonstrated the ability to reduce SRED episodes in a randomized controlled trial compared to a placebo [83]. However, the tolerability of topiramate may pose challenges for its utilization in some patients.

## 4. Sleep Disorders Associated with Eating Disorders: A Cross-Road for Neurologist, Psychiatrist and Nutritionist

### 4.1. Narcolepsy and Eating Disorders

Eating disorders are among the most common disturbances in narcolepsy type I (NT1) patients. NT1 is a rare, autoimmune, life-long disabling, neurological disorder associated with excessive daytime sleepiness, cataplexy and nocturnal sleep disruption, due to the progressive loss of hypothalamic orexinergic neurons [62]. Orexin type A and B play a pivotal role in the regulation of human behaviors in highly motivational situations, including food-related conditions, as they are stimulated by fasting and, vice-versa, inhibited by satiety signals [84]. Physiologically, food choices are under the influence of manifold regulatory systems, not only the ‘homeostatic’ balance, located in the hypothalamus and brainstem, but also the ‘hedonistic’ system, under a cortico-limbic regulation [85]. The orexinergic activity is key in the maintenance of the individual’s metabolic homeostatic balance through its effects on reward-seeking mechanisms and regulating energy balance mechanisms [86]. Finally, orexin also influences the olfactory system, which indirectly modulates appetite threshold, and a role for a mild hyposmia in NT1 in influencing food choices and eating behaviors had been theorized [87]. Besides the typical narcoleptic symptoms, this disorder is characterized by serious metabolic, cardiovascular and psychiatric disturbances that dramatically contribute to worsening the clinical phenotype [88]. NT1 patients are at higher risk for obesity and typically have an increased waist circumference, associated with excess fat storage in abdominal depots [89], a condition which cannot be completely elucidated by their exposure to excessive daytime sleepiness (EDS) (and consequent inactivity), as patients with idiopathic hypersomnia (IS), equally affected by EDS, do not share the same metabolic risk. Narcoleptic patients exhibited higher risk for diabetes mellitus, dyslipidemia and blood hypertension compared to age-matched healthy controls [90].

NT1 is also associated with higher risk for compulsive ‘snacking’ eating, preference for high-calories palatable food, also after satiety, with a behavior similar to the one observed in sleep-deprived animals, leading to the hypothesis of a neural dysregulation of dopamine and opioid systems signaling in the disorder pathogenesis [89]. A strong association between narcolepsy and SRED had also been observed in previous investigations, with 32% of patients affected by NT1 suffering from this parasomnia (vs. 3% of healthy controls) [91].

NT1 frequently overlaps with various psychiatric conditions, such as anxiety, depression or psychosis, which also can promote some dysfunctional eating behavior themselves, overcomplicating patients’ conditions [88].

The literature studies identified that NT1 patients present a higher rate of binge eating, bulimia and/or other compulsive eating behaviors [79]. Furthermore, narcoleptic patients lower metabolic rate compared to age-matched healthy controls. The combination of compulsive eating trends and lower metabolic balance promote a positive energy balance, lastly leading to weight gain and obesity. Therefore, early screening for metabolic alterations and abnormal eating behaviors should be included in the narcolepsy work-up to reduce the morbidity of the conditions and to lower cardiovascular risk of affected patients.

Effect of novel therapies (e.g., Pitolisant, Solriamfetol) with respect to daily and nightly eating behaviors in NT1 patients should be carefully explored, although an intriguing role for the inverse agonist/antagonist H3 receptor (Pitolisant) towards eating behaviors had been theorized [92].

### 4.2. Kleine-Levin Syndrome

‘Abnormal eating behaviors’ are among the cardinal symptoms of Kleine–Levin syndrome (KLS), a rare condition firstly described in 1962 by Critchley and Hoffman [93]. KLS is characterized by recurrent episodes of excessive sleepiness, lasting from few days to four weeks, frequently accompanied by compulsive consumption of large amounts of food (hyperphagia, ‘‘binge eating”) and hypersexuality [62]. It has male predominance and typically occurs during adolescence. It is assumed that the disorder may be a post-infectious auto-immune condition, associated with paroxysmal hypothalamic and or pituitary-hypothalamic axis dysfunction [94]. Other theorized mechanisms involved viral damage to the diencephalic brain areas, anomalies in the serotonininergic/dopaminergic neurotransmission and genetic predisposition. In particular, an association with the *TRANK1* gene locus, previously associated with bipolar disorder and schizophrenia, has recently been demonstrated with genome-wide association study on a large cohort of KLS cases [95,96]. Also, due to observation of a frequent association with human leukocyte antigen (HLA)-DQB1*0201 allele frequency, an autoimmune etiology has also been suggested for this condition [97]. Interestingly, a single case report identified the association of KLS with small-intestine bacterial overgrowth, pointing to the fascinating link between gastrointestinal microbiota and brain function, modulated by either autoimmune or metabolic mechanisms [98]. Benefits may be achieved by the utilization of different medications including stimulating drug, antiepileptic drugs, antidepressants, lithium, cortisone, melatonin, benzodiazepine, antipsychotic and dopaminergic drugs; however, no data on long-term follow-up are currently available and no standardized treatment protocols exist. Abnormal eating behavior in KLS is usually represented by excessive hunger and rapid consumption of large amounts of highly caloric food: the so-called ‘megaphagia’. Weight gain due to compulsive overeating may occur. Coexistent described personality changes include irritability, depersonalization, hypersexuality, depression, confusion and other impulsive behavior. Functional imaging (PET-FDG) data during the acute phase of the condition revealed hypermetabolism in various cortical areas involved in emotional cue processing and sensorial perception [98]. Even if the condition is usually self-limited, it is important for neurologists and psychiatrists to be aware of this diagnosis as it may lead to adverse health (mostly metabolic) consequences. Patients affected by KLS could be misdiagnosed as psychiatric patients, as many or their recurrent symptoms are common among various psychiatric disorders (e.g., unipolar or bipolar depression). In KLS, the association of uncommon disturbances such as excessive daytime sleepiness, amnesia and confusion may suggest the existence of an underlying (subtle) neurological etiology. Notably, although the condition is by definition a self-limited relapsing-remitting disorder, some symptoms could rarely become chronic (e.g., long-term apathy, persistent mild cognitive disturbances, psychological distress), increasing the burden of this pathological condition [98].

### 4.3. Sleep Deprivation and Eating Behaviors

Sleep deprivation characterizes chronic insomnia patients, circadian sleep–wake disorders (CSWDs), shift workers and numerous other sleep-related pathological entities. Recommended sleep time should last ranges from 14 to 17 h in newborns (0–3 months), to 10–13 h in preschool children (3–5 years), 8 to 10 h for teenagers, 7 to 9 h for young adults and 7 to 8 h of sleep for older adults [99]. However, in recent years, nighttime sleep duration has been decreasing worldwide, due to changes in lifestyle, mobile technology and work duties [100]. For instance, the utilization of electronic devices (mobile phones, television, laptop) might affect sleep quality and quantity, yielding to significant increase in sleep latency [100].

Sleep duration regulates various hormone concentrations including leptin, ghrelin, insulin, cortisol, growth hormone and thyroid stimulating hormone [101]. Short sleepers had reduced leptin and elevated ghrelin concentrations: conditions which increase appetite and therefore promote weight gain [102].

A recent systematic literature review attested that short sleep duration is a risk factor for obesity in preschool children (RR: 1.54 [95% CI, 1.33 to 1.77]) [103].

It is clinically common to find insomniac patients, especially those with no education on sleep hygiene, with the habit to eat at night as ‘time-killing’ activity. This unhealthy behavior could have dangerous metabolic consequences and patients should be warned on associated risks.

CSWD, shift-workers and subjects affected by social jetlag may develop weight gain not only as a consequences of sleep loss, but also due to circadian misalignment, a mismatch between endogenous circadian rhythms and behavior, that frequently favor a delay in mealtimes and lastly enhance weight gain and positive metabolic balance [104].

In this framework, the novel concept that *‘a calorie is not always a calorie’* refers to the importance of always focusing on food with a more holistic approach, as many factors differently impact on food’s related health consequences. One of the major determinants is the time when the ‘calorie’ is introduced. Late-night eating, a common behavior in patients with circadian misalignment, increases the risk for adverse metabolic consequences (Figure 3). Furthermore, it has been demonstrated that evenings are indirectly associated with higher food addiction scores mediated by insomnia and impulsivity, whilst morning is usually associated with better control over food [105].

Hence eating behaviors in CSWD may go adrift not only due to wrong meal timing or excessive snacking habits, but also due to higher tendency towards compulsive/addictive behaviors. The vicious cycle thereafter continues due to negative consequences of unhealthy eating habits on sleep quality, exacerbating the initial sleep disorder.

In recent years, intermittent fasting has become popular as a highly effective weight loss method, with significant beneficials on glucose and lipid metabolism [106]. Similarly, time-restricted exercise had been largely adopted to restore metabolic balance. Besides their advantages on the metabolism, tailored training and nutrition timing had been proposed to reset peripheral and central circadian pacemakers, contributing to addressing circadian misalignment [107].

Mechanisms involved in the association between food choices and sleep deprivation are only partially understood. It seems that the selective loss of REM sleep could impact the craving for weight-promoting foods, favoring the assumption of highly palatable foods, and this seems to be mediated by the prefrontal cortex. In fact, in REM-sleep deprived mice, the consumption of highly palatable foods can be inhibited if the prefrontal cortex is chemically inactivated [108]. Indeed, sleep can deeply influence behavioral states, motivation and, specifically, reward processing within the brain and, in parallel, sleep deprivation can lead to severe psychological consequences and induce abnormal reward seeking in affected patients [109].

Additionally, the intricate connection between sleep and diet becomes evident through changes in eating habits caused by what is known as social jetlag. This phenomenon refers to the misalignment between our biological rhythms and social schedules, leading to differences in sleep and activity patterns between free days and work/school days [110]. A recent comprehensive review has demonstrated that social jetlag, affecting about two-thirds of the working-age population [110], is linked to reduced adherence to a healthy dietary pattern [111]. Instead, individuals experiencing social jetlag tend to consume higher amounts of calorie-dense foods, sugar-sweetened beverages and saturated fats.

This intricate relationship forms in the early stages of life and has the potential to evolve alongside the development of various pathologies, including sleep disorders (Figure 4) [112].

From an evolutionary point of view, human beings always preferred highly caloric and energy-rich foods, as they were necessary for survival. Our ancestors had to struggle to find food: a condition distant from the modern era, where food and beverage are easily available [113]. This ‘memory bias’ contributes to the development of unhealthy eating behaviors which, in turn, equate to lower sleep quality.

## 5. Sleep and Microbiota

Gut is considered the ‘second brain’ of the human body. Besides its roles for digestion and nutrients absorption, the gut continuously communicates with the central nervous system in a bi-directional way. This concept gave rise to traditionally known paradigma/expressions such as ‘feeling butterflies in your stomach’ to implicitly describe the gut-to-brain axis. The term ‘microbiota’ refers to the collection of trillions of micro-organisms creating a complex ecosystem mainly constituted by bacteria, together with viruses, fungi, protozoa and archaea. Mechanisms through which gut microbiota may influence brain functioning are not entirely understood.

Among the hypothesized mechanisms it has been suggested that gut-brain communication occurs throughout the autonomic nervous system [114] or the enteric nervous system [115]. Numerous investigations theorized an immuno-mediated/inflammatory-dependent mechanism [116], whilst other focused on the involvement of the hypothalamic–pituitary–adrenal axis. Microbiota may influence neuronal, microglia and astrocytes functioning, and it seems that it may even regulate fetal and adult neurogenesis [117]. A recently published study has showcased that variations in entero-phenotypes exhibit a direct correlation with disparities in inter- and intra-network connectivity within the default mode network, executive control network and sensorimotor systems. This study introduces a new perspective for conceptualizing the relationship between gut and brain functioning [118].

In a broader context, the gut microbiota plays a crucial role in maintaining a robust immune system. Conversely, the presence of intestinal dysbiosis can lead to the development of chronic systemic inflammation, with notable effects on the brain, including an increased vulnerability to neuroinflammatory diseases like multiple sclerosis (MS) [119]. Recently, there has been growing recognition of the significance of the gut–brain axis in various neurological conditions, including strokes [120]. Of particular interest is the role of the microbiota in the progression of neurodegenerative diseases such as Alzheimer’s and PD. Numerous studies have suggested that the intestine and its associated microbiota may contribute to the accumulation of amyloid fibers [120]. Additionally, α-synuclein, when accumulated in the gastrointestinal tract, has the potential to travel to the substantia nigra and striatum through the vagus nerve [121]. This finding implies that the enteric nervous system could play a role in the accumulation of these fibers in the brain. Regarding synucleinopathies like PD and multiple system atrophy, there has been a significant paradigm shift. A hypothesis known as the gut-origin hypothesis has been proposed, based on evidence indicating an ascending spreading pattern of aggregated alpha-synuclein from the gut to the brain [121,122]. This hypothesis is further supported by the observed co-occurrence of inflammatory bowel disease and synucleinopathies [122]. Notably, a recent groundbreaking study demonstrated that the molecular signature of the microbiota in PD is already present in the prodromal stages of the disease and that his association is particularly evident in individuals with REM sleep behavior disorder (RBD) [122].

If the gut can influence brain functioning, triggering several neurological conditions and highly impacting on global cerebral cognition, it might not be surprising that it may impact sleep quality as well. Unhealthy diet reduces sleep quality, while higher adherence to the Mediterranean diet and food quality ameliorates it [123,124]. Thanks to its balanced composition, the Mediterranean diet (MD) is recognized as a key prevention instrument for health conservation. An intriguing and complex role for microbiota has been recently described with respect to various sleep disorders including OSA and insomnia [125] paving the way to novel therapeutic strategies in the field of sleep disorders.

Similarly, gut microbiota had been implicated in eating disorders pathophysiology, including AN and BN. According to this model, AN and BN may originate from an altered signaling between gut microbiota and host immune and neuroendocrine systems, regulating feeding behavior. After exposure to a bacterial antigen similar to the hormone a-MSH, an autoimmune reaction takes place, finally triggering the production of a-MSH cross-reactive autoantibodies, which chronically stimulate the melanocortin system, a key pathway for feeding behaviors regulation [126].

## 6. COVID-19, Sleep and Eating Behaviors

The recently experienced lockdown period, related to COVID-19 outbreak, exerted a massive social and psychological impact worldwide. Abruptly, most people found themselves with much more time to spend at home and to dedicate to cooking activities. In numerous instances, this circumstance has facilitated the emergence of unhealthy eating patterns characterized by excessive consumption of sugary treats, snacks and alcohol [127]. This tendency has been particularly prevalent among individuals who experience anxiety, engage in emotional eating, suffer from depression or have pre-existing eating disorders [128]. Surprisingly, in other cases, people improved their nutrition habits, preferring home-cooked meals, fresh vegetables, legumes and fruits [129]. The quarantine strongly impacted on subjects’ sleeping schedules as well. Interestingly, in most of cases it caused a worsening of sleep quality, correlated with unhealthy eating patterns, once again indicating the existence of a link between the conditions [130,131]. A recently published Italian study explored the impact of the adherence to MD with respect to eating habits during the lockdown period [132]. According to their results, patients subdivided into four clusters: ‘healthy eaters’ (mainly women and elderly with high MD), ‘less eaters’ (mainly men, with low-to-moderate MD), ‘usual eaters’ (prevalently women and elderly, with low adherence to MD) and the ‘more eaters’ (mainly young women with low MD). Similar results had been described by Prete et al. [133]. It seems that adherence to MD improved quality of diet during the lockdown period among Italians. This study indirectly underlined the importance to educate patients towards healthy eating habits in dramatic situations, such as quarantine. The MD is considered among the healthiest dietary pattern worldwide, as it could reduce chronic inflammation, promote metabolic balance and sustain inflammatory system reactivity [134]. Similarly, the MD showed positive effects with respect to numerous sleep disorders including obstructive sleep apnea and insomnia [135,136]. Dietary composition can strongly influence sleep features with different pathways: for example, omega-3 fatty acids modulate the serotoninergic and dopaminergic transmission, high-glycemic index carbohydrates increase the availability of tryptophan for serotonin synthesis and L-ornithin reduces stress-induced activation of the central nervous system mediated by the GABA receptor [137]. By recognizing the risk factors associated with the development of unhealthy dietary habits during the COVID-19 pandemic, and acknowledging the significant link between nutrition and sleep quality, healthcare practitioners and researchers should prioritize these areas when assessing patients during challenging periods like quarantine.

## 7. Impact of Culture and Chronobiology

The tangled relationship between eating behavior and sleep should always be evaluated in a more holistic perspective inclusive of cultural, religious and demographic information, and keep in mind the dynamic evolvement of individual’s chronotype throughout their lifespan. As most living animals, humans require the correct functioning of numerous circadian clocks to adapt properly to environmental changes and to the 24-h planet rotation along its axis. The regulation of inner circadian rhythm is highly complex and dynamically changes with aging, under the influence of hormonal levels, health status and following genetic and epigenetic variations. It has been proved that the human internal rhythm relies on a bidirectional flow between the main circadian pacemaker (suprachiasmatic nucleus) and the numerous peripheral functional clocks, located in almost all tissues and cells [138]. Notably, at the individual level, the circadian rhythm is not uniform over the course of a lifetime, and numerous changes occur starting from the earlier phases after birth, persisting during adolescence and adulthood and continue to evolve with increasing age [139]. Some aspects of chronoregulation and eating habits also appear largely influenced by cultural/religious features: during the Islamic Ramadan, for instance, Muslims abstain from eating and drinking from dawn to dusk and this fasting is considered a pillar of Islam. In Argentina, most people have four meals a day (including breakfast, lunch, afternoon meal, or *merienda*, and dinner), while the Chinese tradition suggest adapting to the introduction of higher and lower carbohydrate meals respecting the transition between the yin (resting phase) and yang (active phase) [140,141]. Equally, it is important to evaluate some cultural features such as the healthy habit for family meals, highly prevalent in Italy and Europe, which have been proved to impact on physical and mental health and seems to prevent numerous maladaptive behaviors [142]. Meal frequency during the day and meal composition can also influence health and energy level, lastly modulating sleep quality and quantity. All these chronobiological aspects largely influence lifestyle, wellbeing and metabolism. The correct alignment between central and peripheral clocks is essential to support metabolic functioning. It is known that morning usually supports healthier dietary habits, reduces the risk for social jetlag and promotes a longer duration of nocturnal sleep in both adolescents and adults, while more evening-type subjects are at higher risk for derailment of both sleep and diet, presenting often unhealthy lifestyle patterns [143,144]. Thereafter, cultural, religious, demographic and circadian features should always be evaluated at an individual level while investigating the relationship between sleep and diet (Figure 5).

## 8. Conclusions

According to Aristotle “sleep occurs when the body element is dragged upwards by the heat through the veins to the head. (…) The animal wakes up when digestion is finished, that is, when the heat that in large quantities had been concentrated by the neighboring regions within a small space prevails and the more full-bodied blood is separated from the purer one’’ (Parva Naturalia, Aristotele, 4th century BC). Modern science has confirmed the close connection between sleep and nutrition and their emotional background appears deeply intermingled. Future research should include their comprehensive evaluation to allow a wider understanding of patients’ health conditions. Careful assessment of nutrition habits and schedule should be included in the routine diagnostic work-up of patients affected by sleep disorders and efforts should be dedicated to their education. Conversely, an accurate description of sleep–wake habits and sleep quality might enrich nutritionists and psychiatric work-up towards patients affected by ED, as they commonly hide their sleeping disturbances. Within this intricate framework, we additionally emphasize the captivating involvement of the orexinergic system and gut microbiota. Cultural, religious and demographic aspects should always be considered in the clinical care of patients with eating and/or sleeping disturbances. These factors have the potential to aid in the phenotyping of sleep and eating disorders, offering a novel, integrated and multidisciplinary perspective.

## Figures and Tables

**Figure 1 nutrients-15-04488-f001:**
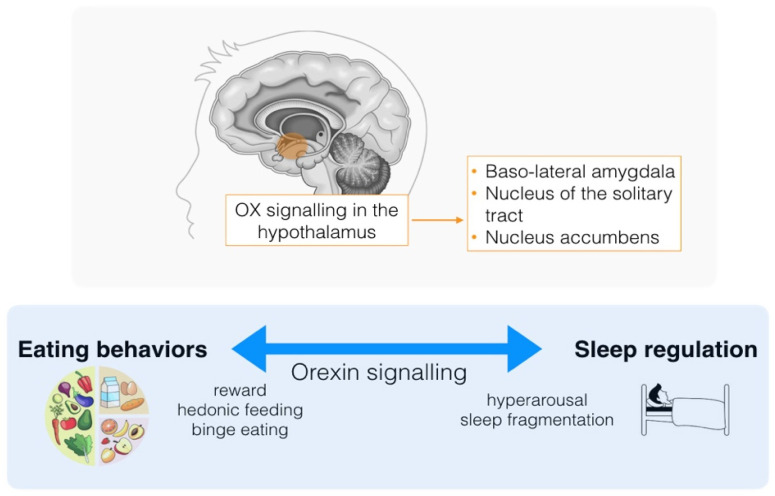
Self-promoting loop between sleep dysregulation and eating behavior mediated by the orexinergic signaling.

**Figure 2 nutrients-15-04488-f002:**
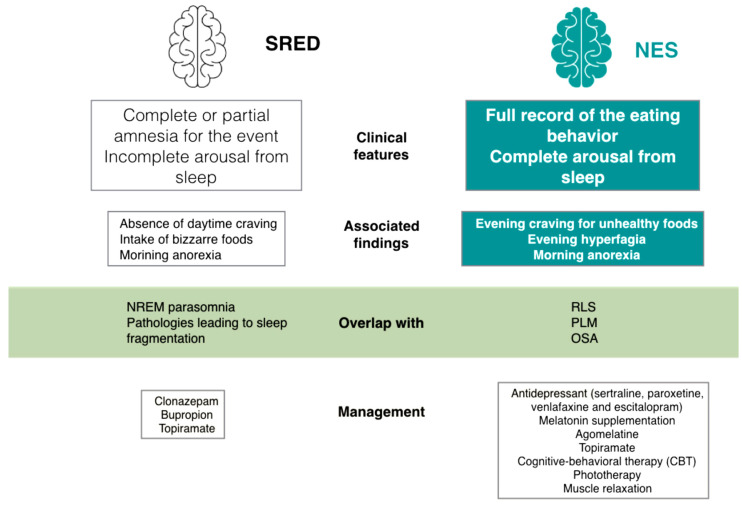
NES and SRED, two distinct sleep-related eating disturbances.

**Figure 3 nutrients-15-04488-f003:**
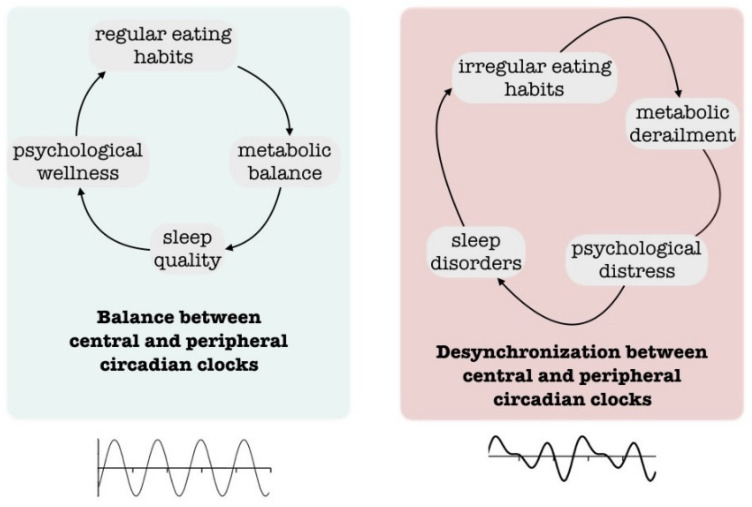
Schematic representation of the relationship between central, peripheral clocks and health.

**Figure 4 nutrients-15-04488-f004:**
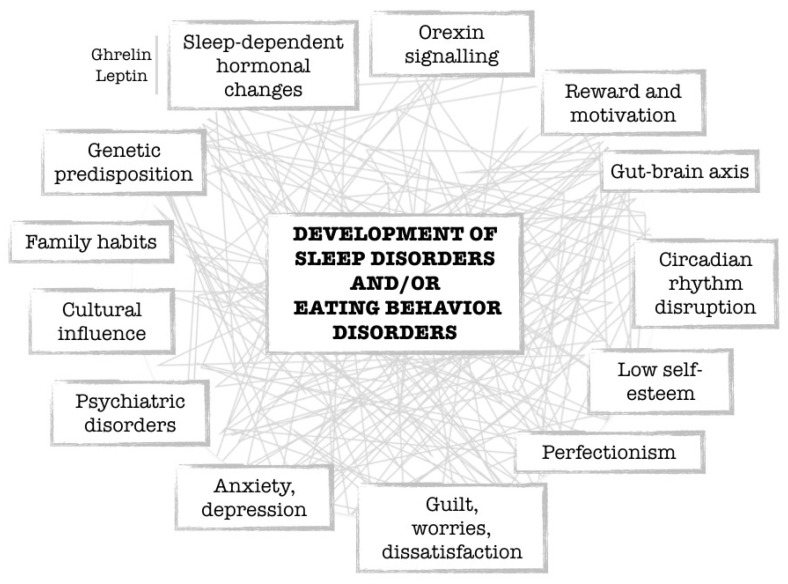
Schematic representation of (overlapping) factors involved in the development of sleep disorders and/or eating behavior disorders.

**Figure 5 nutrients-15-04488-f005:**
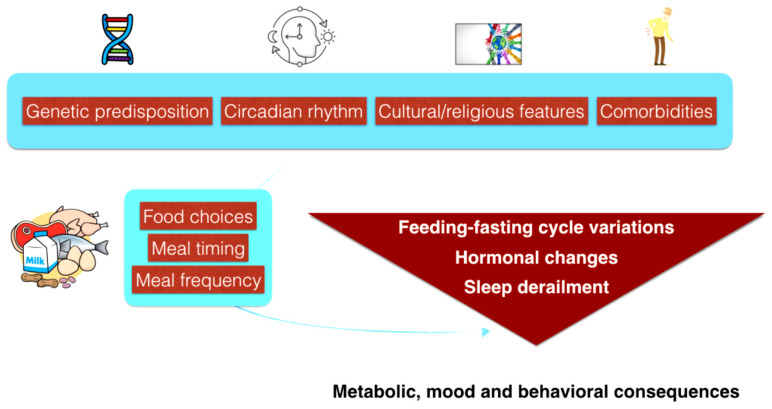
The multi-layered regulation of eating behavior and sleep consequences.

## Data Availability

Not applicable.

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
