# Peer review of "Sleep Pathologies and Eating Disorders: A Crossroad for Neurology, Psychiatry and Nutrition"

_nutrients, 2023, doi:10.3390/nu15204488_

Round 1
Reviewer 1 Report
To identify the process of integrating sources, it is necessary to indicate how the systematic search for information was carried out, including the databases consulted and the keywords used. Likewise, indicate the period in which the information search was conducted, since other sources have been identified that could be relevant for this review and that were not included, probably because they are outside the period in which the review was conducted.
Although the DSM-5 changed the structure of eating disorders, eliminating childhood and adolescent disorders to create a single spectrum that allows diagnosis at any age, factors such as the individual's metabolism, food absorption mechanisms, hormonal status, ethnicity, eating habits according to the availability of food/nutrients, absence/presence of chronic degenerative diseases, age and sex/gender, among others, should be considered to establish the severity or remission of the case. These same factors are associated with some Sleep-Wake Disorders, so it is recommended to integrate them to the review, complementing the information by including genetic liability, hyperarousal, and circadian dysrhythmias. Although some of these factors are mentioned in some sections, it is not consistent throughout the document. Tables could be developed that integrate Sleep Pathologies, Eating Disorders, some of the factors mentioned and the 3 areas: Neurology, Psychiatry and Nutrition.
The section on Bulimia Nervosa presents little information compared to the other sections.
Figure 1. Self-promoting loop between sleep dysregulation and eating behavior mediated by the orexinergic signaling: The schematic representation of the figure is very general, considering that the explanation in the text is more extensive. It could be improved if it is presented integrating Sleep Pathologies and Eating Disorders.
The article is interesting, however, the way in which the information is presented, and the organization of the information does not meet the objective of the paper. The key is in the integration of the information.
Author Response
We would like to thank the anonymous reviewer for his/her insightful comments.
It is indeed absolutely important to evaluate the contribution of many confounding factors such as hormonal status, chronobiology, eating habits or cultural/religious features when evaluating sleep and eating behaviors.
We add an entire new paragraph in the end of the text highlighting this key observation, which we believe greatly improved our work.
Many thanks for this precious suggestion.
Also, we added two new figures that we hope could help the reader following the text and which aim to partially integrate the informations provided in the text.
Finally, we added some further notions regarding PSG data of patients affected by bulimia nervosa in the dedicated paragraph, as suggested.
Reviewer 2 Report
This review presented the connections between sleep and eating disorders from perspective of nutrition, psychology etc. I would have some comments:
1. the authors started the introduction section since the historical development of human beings, I would wonder is it helpful to illustrate the aim of the review, it may be better if the evidence retrieved to modern life. What the authors think?
2. As far as I know, eating behaviours are diverse from continent to continent, from country to country. for instance, nocturnal eating syndrome, in asian countries like China or Korea, there is a culture for food hunting during night, which is like social activity and entertainment.
3. The review provided comprehensive review of eating disorder and sleep, with a wide range of evidence. My concern would be how the conclusion convince readers?
The manuscript reads well.
Author Response
Many thanks to the anonymous reviewer for his/her important comments.
We specify in the introduction that in this review we focused on the modern era, as suggested.
We added a new entire paragraph on the key role for cultural/religious and chronobiological features in determining differences in terms of sleep/eating behaviors. Many thanks for this absolutely important suggestion, we believe this new paragraph enriches our work.
Reviewer 3 Report
Dear Authors
I have read your paper with great interest. The article is a review that summarize the key aspects of the intermingled relationship between sleep disorders and eating disorders combining a clinical and psychological perspective.
You have observed and properly described that sleep disturbances are a common and persistent manifestation of altered eating behaviors that affects a large proportion of the population and deserves careful monitoring. The description of the different eating habits and eating disorders as well as sleep disorders are sound, clear and require no further improvements.
The Conclusions are correct.
Comprehensive references information is given. However, an important cite is missing to explain that EDS and overweithg are linked, from alterations in hormones like ghrelin and leptin to the hypothalamic-pituitary-adrenal (HPA) axis and pro-inflammatory cytokines. Vgontzas el al. Ann. NN Y Acad Sci 2006; 1083:329-344; doi 10.1196/annals.1367.023
In addition, two important cites should be added to the Kleine-Levin section:
Dauvilliers Y et al. Neurology 2002; doi 10.1212/01
Ambati A et al. Proc Natl Acad Sci USA 2021; doi 10.1073/pnas.2005753118
The Figures are clear and informative.
I agree that prospective studies and research are needed to eveluate how these conditions will change over time; it is obvious they affect a large group of population.
I have no further remarks.
I
Author Response
Many thanks to the anonymous reviewer for her/his encouraging comments, we do really appreciate that! Following his/her important suggestion we added the missing references in the text. Many thanks for your comments.
Reviewer 4 Report
Authors reviewed “Sleep Pathologies and Eating Disorders: A Crossroad for Neurology, Psychiatry and Nutrition, the write up is very clear and written precisely. However, there is need of following changes and should be improve more.
In introduction section, “Moving from arboreal to terrestrial environment……… increased the proportion REM sleep periods” authors should add citation.
Since in this review focused on sleep and eating disorders, authors should explain more in details and add more figures about the neuronal circuits that regulate the sleep and eating behaviors.
“This epidemiological overlap between sleep disturbances and ED ………………. in orexin, cortisol, ghrelin and leptin signaling” citations needed.
The author should write about Prader-Willi Syndrome (PWS) and include sleep and eating abnormalities in this syndrome.
Thanks
Author Response
Many thanks to the anonymous reviewer for his/her comments and suggestions.
We added the missing reference in the text, also we developed two new figures, as suggested. We also added a new paragraph on obesity and sleep disturbances, where we provided some clinical information on Prader-Willi Syndrome as well. Many thanks for your important notes.
Round 2
Reviewer 1 Report
The authors addressed the observations made.